# Recent Progress and Future Opportunities for Optical Manipulation in Halide Perovskite Photodetectors

**DOI:** 10.3390/nano15110816

**Published:** 2025-05-28

**Authors:** Jiarui Zhang, Chi Ma

**Affiliations:** Department of Optoelectronic Information of Science and Engineering, School of Science, Jiangsu University of Science and Technology, Zhenjiang 212100, China; jiarui_zhang1@stu.just.edu.cn

**Keywords:** halide perovskite photodetectors, local surface plasmon resonance (LSPR), surface plasmon polaritons (SPPs), optical nanostructures, special light detection

## Abstract

Perovskite, as a promising class of photodetection material, demonstrates considerable potential in replacing conventional bulk light-detection materials such as silicon, III–V, or II–VI compound semiconductors and has been widely applied in various special light detection. Relying solely on the intrinsic photoelectric properties of perovskite gradually fails to meet the evolving requirements attributed to the escalating demand for low-cost, lightweight, flexible, and highly integrated photodetection. Direct manipulation of electrons and photons with differentiation of local electronic field through predesigned optical nanostructures is a promising strategy to reinforce the detectivity. This review provides a concise overview of the optical manipulation strategy in perovskite photodetector through various optical nanostructures, such as isolated metallic nanoparticles and continuous metallic gratings. Furthermore, the special light detection techniques involving more intricate nanostructure designs have been summarized and discussed. Reviewing these optical manipulation strategies could be beneficial to the next design of perovskite photodetector with high performance and special light recognition.

## 1. Introduction

Photodetectors (PD), which have been widely applied in fields such as optical communications, radiation detection, biological/chemical analysis, and environmental surveillance, can directly convert weak optical signals into distinguishable electrical signals (current or voltage) [1,2,3]. The most fundamental working mechanism in photodetectors is the generation and extraction of photo-induced carriers, which can be further refined as the generation and separation of electron-hole (e-h) pairs after the absorption of photons and the transportation and extraction to external circuitry perform as an observable current or voltage. Photodetectors based on traditional photo-responsive materials, such as silicon for ultraviolet-visible detection and III–V or II–VI compounds for infrared detection, have undergone significant limitations in their application to flexible and wearable devices attributed to high preparation temperatures, stringent operating conditions, and inherent fragility of these materials [4,5,6,7]. Moreover, in low-dimensional systems, one-dimensional semiconductor nanowires (1D) can provide very large surface-to-volume ratios and directional charge transport, and two-dimensional quantum-well heterostructures exhibit tightly bound excitons whose energies can be tuned by layer composition [8,9,10]. These nanostructured PDs can achieve high performance but generally require complex epitaxial growth or assembly and remain challenging to integrate into flexible devices. Therefore, numerous new photosensitive materials have been investigated to satisfy the demands of miniaturization, high quantum efficiency, and low power consumption in the new generation of optoelectronic devices [11,12,13,14].

Among these burgeoning detection materials, the perovskite materials with adjustable direct bandgap, low-cost preparation processes, high absorption coefficients, and long carrier diffusion lengths have attracted considerable attention in the field of optoelectronics. The chemical formula of halide perovskites is ABX_3_, where A denotes the monovalent cation (~Cs^+^, Rb^+^, CH_3_NH_3_^+^ or HC(NH_2_)_2_^+^), B represents the bivalent metal cation (~Pb^2+^, Sn^2+^ or Ge^2+^), and X embodies the halide anion (~Cl^−^, Br^−^, I^−^ or their mixtures) (Figure 1a). Moreover, it can be composed with an octahedron grid frame in 3D space and presented as an equiaxed crystal structure [15,16,17,18,19,20]. Furthermore, the tolerance factor t, defined as $t=(R_{A}+R_{B})/\surd 2(R_{A}+R_{B})$, where the R_A_, R_B_, and R_X_ indicate the ionic radii of the ions at the A, B, and X sites, respectively, has been presented since 1926 to quantitative evaluate the structural stability with different composition of ion sizes. The tolerance factor of stable material should be located between 0.78 and 1.05. Some representative halide perovskite compositions and their detection wavelength ranges are summarized in Figure 1b. By adjusting the stoichiometric ratio of ions, various types of perovskites (2D or 3D) can be easily fabricated and effectively applied in numerous types of modern photodetectors [21,22,23,24]. For example, Li et al. have proposed a 2D-(graded 3D) perovskite heterojunction device under a pyroelectricity-based acceleration strategy, and this device achieved a record high response speed (27.7 ns with an active area of 9 mm^2^) and responsivity (0.65 A W^−1^) under zero bias [25]. Fang et al. have integrated an amorphous organic Rubrene film into a CsPbBr_3_-based photodetector and presented device stability (maintaining 90% original performance after being immersed in water for 6000 min) [26].

In recent years, there have been significant advancements in the field of perovskite photodetectors; however, the basic optimization mechanism still needs to be further explored. Particularly, the optimization of photoelectric performance and enhancement of stability are perennial challenges in this field. Additionally, the optimization strategies primarily involve the following two types. First is inside adjustment with stoichiometric ratio optimization, ion substitution, and crystallization process adjustment [27,28,29,30,31,32]. Recently, Gao et al. have applied synergistic composition engineering with the incorporation of GA (Guanidinium, a strong organic cation with the formula C(NH_2_)^3+^) on A site and the doping of alkaline-earth metal on B site, and successfully solved the trade-off between detection performance ((2.6 ± 0.1) × 10^4^  μC Gy_air_^−1^ cm^−2^ under 1  V cm^−1^ (μC Gy_air_^−1^ represents the charge generated by the detector when the air absorbed dose is 1 Gray = 1 J/kg)) and device stability (beyond half a year) [33]. The second type refers to the outside adjustment with the assistance of additives and encapsulation materials (~polymers, graphene, or even 2D perovskites). The incorporation of additives, including but not limited to thiocyanate ions, diammonium molecules, metal oxides, and metal salts, all aim at the defect passivation on the surface or in bulk, contributing to reducing the recombination losses and optimizing the charge transportation [29,34,35,36]. Huang et al. have utilized ammonium bromide to simultaneously passivate the deep trap defects of FA (Formamidinium, a cationic organic compound with the formula HC(NH_2_)^2+^) vacancies, uncoordinated lead, and Pb-Pb dimers in FAPbBr_3_ while effectively lessening the dark current by 10 times [37]. In summary, both internal and external optimization strategies predominantly focus on mitigating lattice mismatch. Nevertheless, a significant limitation arises from the restricted applicability of each method. The primary determinant lies in their fundamental intervention mechanism: the modification of crystallization through chemical bonding.

In addition to these chemical regulatory schemes, several purely physical strategies have also been proposed, presenting the advantages of local surface plasma resonances (LSPRs), surface plasmonic polaritons (SPPs), and Tamm plasmons [38,39,40]. The present review provides a comprehensive review of the performance optimization of perovskite photodetectors through optical manipulation strategies (Figure 2). The optical nanostructures with periodic and aperiodic configurations, as well as the incorporation method of adding materials or directly fabricating onto the perovskite layer, have been completely summarized and analyzed. Furthermore, the potential advancements in physical optimization mechanisms are discussed, highlighting their advantages in special light detection.

Photodetectors operate based on the photoelectronic effect, and their device structures mainly consist of photodiodes, photoconductors, and phototransistors [41,42,43]. The device performance can be characterized by parameters such as responsivity, quantum efficiency, response time, bandwidth, noise, and specific detectivity [44]. Given the extensive coverage of the relevant introduction in previous literature, the discussion is not reiterated here; instead, the focus is directly on exploring the device performance manipulation of diverse optical nanostructures. With the advancement of device miniaturization, achieving high light absorption efficiency becomes increasingly crucial owing to the continuous thinning of the active layer. Consequently, optical manipulations through local surface plasma resonances (LSPRs), surface plasmonic polaritons (SPPs), resonant cavities, photonic crystals, and other techniques have attracted significant attention [45,46,47,48].

## 2. Disperse Optical Nanostructures

Optical manipulation utilizing dispersed nanostructures typically relies on the phenomenon of local surface plasmon resonances (LSPRs) from plasmonic nanoparticles (PNPs). LSPRs refer to the coherent collective oscillation of electrons in metal nanostructures and occur when the incident light frequency matches the natural oscillation frequency of free electrons within the metal nanostructure [49]. As a result, a tens to thousands of times greater electric field appeared around the PNP compared to the incident light. This significantly enhanced field decays quickly as it moves away from the PNP surface, generating the so-called near-field enhancement phenomenon [50]. However, it should be noted that metal nanoparticles also absorb a portion of the incident light and convert it into heat; excessive photothermal heating can raise the local temperature and potentially damage the perovskite, harming device performance [17,51]. It is also important to note that if the metal nanoparticles are very close to the perovskite absorber or emitter (≤10 nm), their near-field will quench the excited state in a non-radiative manner [52]. In addition, there is a lower size limit below which LSPR effects weaken: very small particles experience increased electron–surface scattering and resonance damping, resulting in broadened and diminished LSPR. The field enhancement from a plasmonic NP is directional and depends on its orientation and position relative to the active layer [53]. The specific resonance wavelengths can be easily manipulated by adjusting the sizes, shapes, compositions of the metal nanostructure, and the dielectric of surrounding material [54]. Optimization of perovskite photodetectors based on plasmonic phenomena involves four distinct mechanisms: light scattering, plasmon resonant energy transfer, hot electron transfer, and electromagnetic fields categorized as radiative effects (Figure 3a) [55]. When the size of PNPs is significantly smaller than the incident light wavelength, the enhancement of LSPRs is observed through either relaxation and re-radiation of light into the active layer or by acting as a secondary light source to amplify local electric fields (radiative effect) [56,57]. Alternatively, the energy can transfer into the adjacent semiconductor to enhance current generation (non-radiative effect) [58].

Gold is the preferred material for fabricating the desired optical nanostructures because of its inherent chemical stability and exceptional biocompatibility. For example, Peng et al. applied the gold nanoparticles (AuNPs) to enhance light–matter interaction (Figure 3b) and employed photon manipulation to simultaneously achieve higher photosensitivity (~10^5^), lower detection limit (5.7 pW), and wide spectral response (UV-vis-NIR) [59]. The upconversion phenomenon, which indicates the conversion of photons in a lower energy state into a higher energy state, has been currently proposed to enhance the detectivity of light in infrared wavelengths. Ko et al. combined a polymeric microlens array (MLA) film and a hierarchical plasmonic upconversion (HPU) film into one hybrid plasmonic upconversion architecture to effectively utilize the local field amplification in the proximity of AuNPs (Figure 3c). Different from the traditional Lanthanide-doped upconversion nanoparticles, the newly proposed hierarchical metal nanostructures are composed of a core-satellite nano-assembly architecture, with the adsorbing of 15 nm AuNPs on a 150 nm AuNPs monolayer (Figure 3d). Moreover, the absorption wavelength can effectively red-shift to 1550 nm. Consequently, a triple-cation perovskite photodetector with the assistance of this structure yields a superior responsivity and detectivity of 9.80 A W^−1^ and 8.22 × 10^12^ Jones (1 Jones = 1 cm·Hz^1/2^/W), respectively, under a weak power density of ≈0.03 mW cm^−2^ [60]. In addition to their spherical structure, the Au nanoroads (AuNRs) can serve as nanostructures for optical manipulation. Kin et al. integrated the AuNRs into a vertically structured methylammonium lead triiodide (CH_3_NH_3_PbI_3_) photodetector, achieving a significant enhancement of photocurrent and an impressive responsivity of 320 A W^−1^ at a low driving voltage of −1 V [61].

The perovskites exhibit relatively weak coupling efficiency within the infrared wavelength range, leading to a deteriorated performance of perovskite-based photodetectors. In addition to modifying the sizes of metallic spheres or nanorods, directly fabricating discrete structural arrays can obtain precise manipulation of light with specific wavelengths. Fang et al. proposed a multilayer plasmonic-functionalized substrate with the construction of Au nano-squares arrays/SiO_2_ spacer/Au film (Figure 3e). This design generates a significant localized electric field around the Au nanostructure, bringing about enhanced photocurrent within the visible/near-infrared range and an outstanding maximum external quantum efficiency of 65% [62]. Similarly, the Au bowtie nanoantenna (BNA) arrays have been incorporated into a triple-cation mixed metal halide perovskite photodetector by Guo et al. This plasmonic modification revealed a huge enhancement factor of 2962% at a light wavelength of 785 nm. Although the highly ordered discrete structure enables precise adjustment of specific wavelengths, adjustment in large areas remains challenging due to limitations in preparation costs [63].

**Figure 3 nanomaterials-15-00816-f003:**
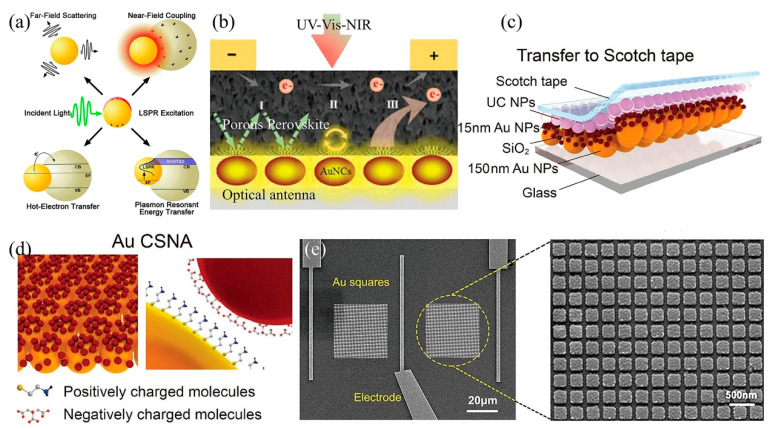
Schematic illustration of (**a**) Plasmon enhancement mechanisms of radiative effects. Copyright permissions from Reference [55]. (**b**) Device design for nanogold-perovskite hetero-structured photodetector. Copyright permissions from Reference [59]. (**c**) Hybrid plasmonic upconversion architecture; (**d**) Hierarchical metal nanostructures. Copyright permissions from Reference [57]. (**e**) SEM images of the plasmonic substrate at different magnifications. Copyright permissions from Reference [62].

Silver or aluminum-based optical nanostructures have also been extensively applied to mitigate the cost of preparation [18,64]. The enhancement of light harvesting efficiency is primarily determined by the localized surface plasmon resonances induced antenna-amplified light scattering and local electric field enhancement [65,66]. A detailed repetition of the related work is omitted. The transportation of photogenerated charges inside the device should be fully considered, even though the significantly enhanced evanescent electromagnetic field can optimize the photoelectric conversion efficiency in the photosensitizer layer. For instance, the optimization strategies commonly involve the integration of metallic nanoparticles into the photoactive materials to establish direct contact [67]. This optimization strategy inevitably gives rise to several drawbacks that compromise the efficiency of charge transportation and extraction, including the photogenerated charge carrier recombination induced by surface defect sites on metallic nanoparticles, as well as the phenomenon of surface energy transfer (Figure 4a). Additionally, the heat stemming from the decay of LSPRs stimulates structural and chemical changes in perovskite materials and then degrades the device’s performance. Liu et al. proposed an AgNPs-embedded SiO_2_ composite optical manipulation strategy to avail LSPRs enhancement while mitigating unfavorable phenomena (such as charge, energy, and heat transfers) (Figure 4b). Finally, a significant enhancement factor of 7.45 in responsivity was achieved through the integration of plasmonic nanoparticles into perovskite photodetectors [68]. Based on similar optimization principles, Lei et al. reached a harmonious balance between plasmonic near-field enhancement and surface energy quenching by integrating an ultrathin Al_2_O_3_ spacer onto a silver nanoparticle membrane (Figure 4c). This proposed strategy ultimately yielded a 6.5-fold enhancement in photocurrent and a detectivity of 4.27 × 10^11^ Jones at 410 nm within a lead-free perovskite quantum dots photodetector. Furthermore, they discussed the relationship between the nonradiative quenching (through surface energy transfer) and plasmonic near-field enhancement (Figure 4d). The amplification of electric field intensity through spacer thickness reduction and the quenching effect governed by a 1/d^4^ dependence (where d represents the spacer thickness) should be considered simultaneously for optimal plasmon-related enhancement [69,70]. Some hybrid optimization strategies have also been proposed for manipulating incident light. Wu et al. decorated the metal nanoparticles onto anodic aluminum oxide (AAO) to form a hybrid plasmonic nanostructure. By adjusting the nanoparticle and pore sizes, the photo-response behavior can be readily enhanced for specific wavelengths (Figure 4e) [71].

**Figure 4 nanomaterials-15-00816-f004:**
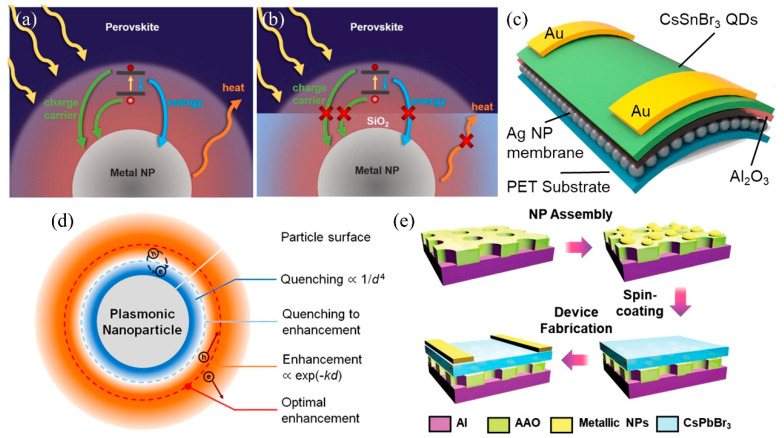
Schematic diagrams of a metal NP embedded in a perovskite thin film with (**a**) and without (**b**) direct contact with perovskite. Copyright permissions from Reference [67]. (**c**) Schematics of the flexible photodetector based on a PET/Ag NP/Al_2_O_3_/CsSnBr_3_ QD structure. Copyright permissions from Reference [71]. (**d**) Schematic of the competing plasmonic near-field enhancement and surface energy quenching mechanisms in a perovskite QD near an Ag NP. Copyright permissions from Reference [68]. (**e**) Schematic fabrication of the hybrid plasmonic nanostructure-based photodetector. Copyright permissions from Reference [71].

As previously discussed, metal nanoparticle-based light management generally relies on coherent electron oscillations to strengthen the light–matter interaction, functioning as a light absorber or an element for subwavelength light confinement. However, the nonradiative plasmon decay process (SPs rapidly dephase and transfer the energy to excite a single electron), which is considered a plagued loss in practical applications, is useful for device optimization. When incident light illuminates a metal nanoparticle, a portion of the photons undergoes scattering phenomena, while another portion whose energy is higher than the bandgap energy (Eg) is absorbed and excites electrons to a higher energy level (leaving a positive charge behind in the valance band termed as an excited hole). If the excitation energy is far higher than the bandgap energy, the jumped charge carriers with even higher energy states are hot electrons (holes), collectively called hot carriers (Figure 5a) [72].

Although numerous strategies have been employed to mitigate the nonradiative SP decay, its inherent parasitic nature renders it an inevitable phenomenon. Therefore, some researchers have begun to accept these imperfections and try to turn waste into wealth. Zeng et al. integrated the optical nanostructure (AuNPs) into a 2D inorganic cesium lead halide perovskite photodetector and significantly optimized carrier transmission by leveraging plasmonic-enhanced light absorption (Figure 5b) and the hot-electron effect (Figure 5c). This yielded the simultaneous achievement of a wide linear dynamic range (LDR) of 120 dB and a 20-fold enhancement in external quantum efficiency [73]. Yao et al. incorporated the poly (3-aminothiophenol)-coated gold (Au-PAT) nanoparticle into polycrystalline organic-inorganic halide perovskite films to motivate plasmon excitation and then fill the deep-level traps at grain boundaries. Specifically, the hot electrons generated from plasmonic gold nanoparticles can be injected into the adjacent perovskite material across the PAT cover layer, occupying the deep traps at grain boundaries and presenting as electrically neutral in total (Figure 5d) [74]. In addition to utilizing the hot electrons, harnessing the plasmonic thermal effect holds potential for optimizing perovskite materials. Lei et al. addressed the stability issue arising from residual strain by incorporating SiO_2_-coated gold nanorods to achieve in situ strain relaxation through plasmonic local heating. The controlled thermal energy from LSPR decay softens the perovskite lattice, enabling defect annihilation and grain boundary reconstruction without bulk degradation (Figure 5e) [75].

**Figure 5 nanomaterials-15-00816-f005:**
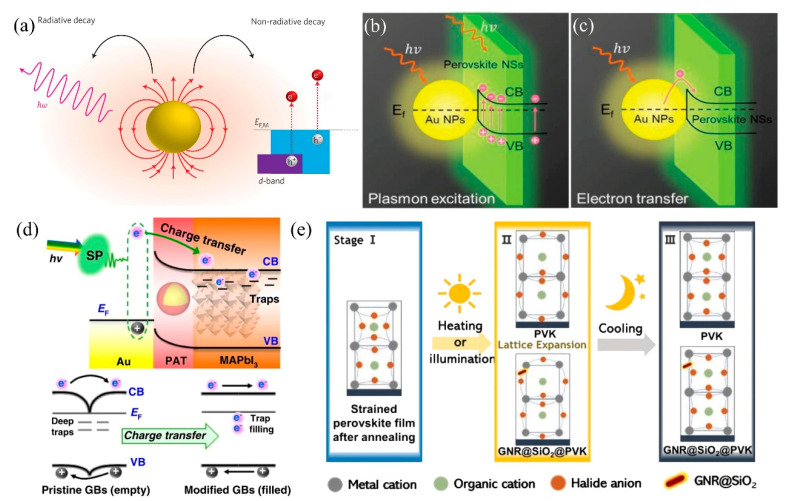
(**a**) Localized surface plasmons can decay radiatively via re-emitted photons or non-radiatively via excitation of hot electrons. In noble-metal nanostructures, non-radiative decay can occur through intra-band excitations within the conduction band or through inter-band excitations resulting from transitions between other bands (for example, d bands) and the conduction band. Copyright permissions from Reference [72]. Schematic diagram of plasmonic enhancement of light absorption (**b**) and Hot-electron effect (**c**). Copyright permissions from Reference [73]. (**d**) Schematic diagram of the PHET mechanism at Au-PAT/MAPbI_3_ interfaces for trap filling. Copyright permissions from Reference [74]. (**e**) Schematic diagram of the mechanism of local heating-induced strain release by GNR@SiO_2_ incorporation. Copyright permissions from Reference [75].

## 3. Periodic Optical Nanostructures

In contrast to the utilization of local surface plasmon resonance (LSPRs), optical nanostructures with precisely distributed periodic characteristics predominantly rely on surface plasma polaritons (SPPs). Unlike the confinement of LSPRs close to specific optical nanostructures, SPPs occurring at the metal and organic interface propagate along the surface and exhibit strong binding to the associated interface within a skin depth of tens of nanometers (Figure 6a) [76,77,78]. The SPPs can be effectively manipulated at specific wavelengths by manipulating the metallic materials, incorporating periodic metal electrodes, or directly structuring perovskite materials.

In the initial phase, the periodic optical nanostructures are typically integrated by structured metal electrodes. Therefore, researchers are making efforts to develop cost-effective and simplified strategies for fabricating optical structures at the nanoscale (even below 10 nm) with controllable dimensions. Some newly developed technologies, including nanoimprinting, electron beams, scanning probe lithography, and extreme ultraviolet, have been proposed to fabricate nanostructures onto photoresist and transfer them to metallic film [79,80]. Vörös et al. introduced a template-stripping-based nano-transfer printing method and transferred arbitrary thin film metal structures onto a variety of substrates (such as PDMS, Kapton, silicon, and glass) (Figure 6b) [81]. The major advantage of this technique lies in its exceptional compatibility with the roll-to-roll fabrication process and even being suitable for curved substrates. Considering that the optical manipulation facilitated by these meticulously engineered metal nanostructures has been extensively discussed in previous reviews, further elaboration is obviated [82,83,84,85]. Additionally, the pre-deposited structured substrate inevitably exerts an influence on the perovskite crystallization process, leading to a degradation in device performance. In the present review, the strategies for direct construction of optical nanostructures onto perovskite films are discussed, enabling simultaneous realization of nanostructure fabrication and crystallization manipulation.

A few years ago, Hu et al. reported a high-performance perovskite photodetector with nano-patterned structures, as well as improved crystallinity after the fabrication of a nanoscale pattern onto their perovskite film (Figure 6c) [86]. The device exhibits a 35-fold enhancement in responsivity and a significant 7-fold improvement in the on/off ratio. This optimization strategy is based on the high-temperature (100 °C) and high-pressure (7 MPa) healing process. During the nanoimprint, the application of elevated pressure and temperature facilitates the migration and coalescence of small grains toward larger ones within the confined mode cavity, contributing to the optimized grains and defects (dislocations, disclinations, and vacancies). Subsequently, Park et al. adopted a polyurethane acrylate (PUA) template for the fabrication of vertically grown halide perovskite (VGHP) nanopillar photodetectors through the nanoimprint crystallization technique. The optimized VGHP films exhibited remarkably curtailed defect density and enhanced charge transport capability [87]. The nanoimprint strategy can also be employed to fabricate a CsPbBr_3_ perovskite nanowire array, which was then integrated with a conjugated polymer to form a hybrid structure for the development of a self-powered photodetector (Figure 6d) [88]. Additionally, the commercially available DVD-R master was utilized as a substitute for the conventional Si or polydimethylsiloxane master to circumvent the limitations associated with fragility and time consumption. Finally, the device performance demonstrated tremendous enhancement with a broad response spectrum (300–950 nm), high responsivity (0.25 A W^−1^), large detectivity (1.2 × 10^13^ Jones), and fast response speed (111/306 µs).

The model-assisted high-temperature and high-pressure extrusion process was further optimized in the subsequent investigation. To manage the challenges of uncontrolled rapid crystallization and environmental instability of structured perovskite, Park et al. established a highly effective approach by combining soft nanoimprint and a moldable poly (ethylene oxide) (PEO) adduct to fabricate an environmentally stable nanopattern of inorganic halide perovskite (IHP) and demonstrate its potential for advanced photoelectronic applications (Figure 6e) [89]. The well-defined IHP nanopattern (CsPbBr_3_ or CsPbI_3_) was achieved with a width of 200 nm over a large area through the utilization of a topographically prepatterned elastomeric mold imprinting technique. The subsequent polymer backfilling process involves the thermal melting of poly (vinylidene fluoride-co-trifluoroethylene) (PVDF-TrFE), forming air-exposed perovskite nanopatterns that were laterally confined with a thin PVDF-TrFE film. These patterns exhibited exceptional environmental stability for over one year under ambient conditions.

**Figure 6 nanomaterials-15-00816-f006:**
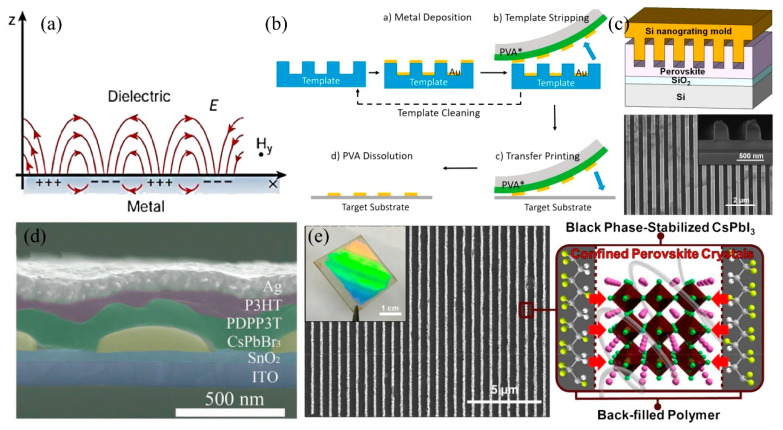
(**a**) SPPs at the interface between a metal and a dielectric material have a combined electromagnetic wave and surface charge character. Copyright permissions from Reference [76]. (**b**) The process flow of template-stripping-based nano-transfer printing. Copyright permissions from Reference [81]. (**c**) Schematic diagram of the NIL process with Si nanograting mold and scanning electron micrograph of nonimprinted perovskite thin film. Copyright permissions from Reference [86]. (**d**) SEM images of P3HT-PDPP3T bulk heterojunction/CsPbBr_3_/SnO_2_ structure. Copyright permissions from Reference [88]. (**e**) FE-SEM image of the line nanopattern of CsPbI_3_ processed by PAN−PA on a large scale with the surface fringe pattern (inset photograph). Copyright permissions from Reference [89].

In response to the growing demand for highly efficient light utilization in thin perovskite films, an increasing number of optical nanostructures have been devised to handle challenges such as strong Fresnel reflection, transmission loss, and low light absorption efficiency [90]. These designs and constructions gradually change from single structures to composite structures. Specifically, the dual interface grating gradually replaces the single optical nanostructure dots or grating, combining both the front and back interface optimization for broadband absorption enhancement. Nonetheless, the absorption enhancement effect of simple parallelly arranged dual interface gratings strongly relies on the precise lateral positioning of the two gratings, imposing challenging alignment tolerances that impede their practical implementation. Li et al. presented an efficient integration of Moiré nanostructures with perovskite photodetectors to achieve a highly sensitive digital polarization imaging by designing a stacked dual shallow grating structure (Figure 7a) [91]. The Moiré interference structure exhibits exceptional light management capabilities by increasing high-order diffraction channels and naturally elongating optical paths, thereby lessening reflection loss. This provides advantages over single and independent light-trapping structures. Furthermore, the hierarchical nanostructure facilitates charge extraction while minimizing surface recombination and shunt current attributed to its large surface area and suppressed defects and voids at the interface during micro-nano imprinting. Finally, the synergistic effects of feedback reflection, diffraction, and extraction of waveguide modes contribute to exceptional device performance, characterized by high responsivity (15.62 A W^−1^), detectivity (5.58 × 10^13^ Jones), and on/off ratio (2.7 × 10^4^).

Subsequently, Sun et al. designed an innovative approach to fabricate in situ encapsulated Moiré lattice perovskite photodetectors and directly achieved the construction through two nanograting-structured soft templates with varying relative rotation angles (Figure 7b) [92]. The prepared soft-structured template effectively restricts the growth of perovskite and facilitates further optimization of crystallization during the formation of the Moiré lattice structure. With the Moiré structure, an impressive responsivity of 1026.5 A W^−1^ is achieved, and the structured perovskite photodetector demonstrates impressive long-term stability with 95% retention of its initial performance after 223 days. The incorporation of specialized optical nanostructures not only enhances light absorption but also enables specific applications, particularly in the realm of sensitive polarized light detection. As the polarization angle of incident light gradually changes, the photocurrent consistently changes from a maximum value (I_max_) to a minimum value (I_min_). This optimized Moiré-lattice perovskite PD can provide a high anisotropy ratio (I_max_/I_min_ = 9.1) (Figure 7c,d). Notably, the Moiré pattern formation can be considerably altered by varying rotation angles (Figure 7e) [93]. In addition to grating widths and periods, the optical properties can be manipulated with greater flexibility and precision.

**Figure 7 nanomaterials-15-00816-f007:**
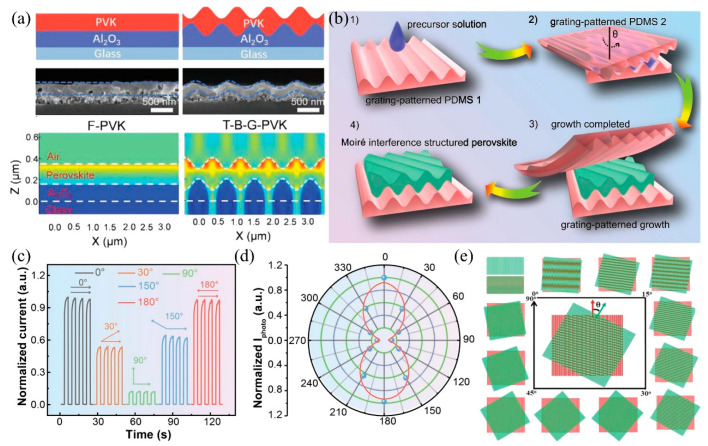
(**a**) Cross-sectional SEM images and field plots of time-averaged electromagnetic energy density with respect to the x–z plane at the wavelength of 650 nm of F-PVK and T-B-G-PVK films. Copyright permissions from Reference [88]. (**b**) The schematic diagram of the fabrication of perovskite crystals with Moiré lattice microstructure. Copyright permissions from Reference [92]. (**c**) I–t curves of Moiré lattice perovskite PD under illumination at different polarization angles. Copyright permissions from Reference [92]. (**d**) Photocurrents of a Moiré lattice perovskite PD under different polarized lights. Copyright permissions from Reference [92]. (**e**) Preparation of Moiré structure by rotating the two diffraction gratings from 0° to 90°. The Moiré period depends on the rotation angle between the two gratings. Copyright permissions from Reference [93].

## 4. Polarization Sensitive Photodetector

The incorporation of optical nanostructures into perovskite photodetectors can significantly enhance the efficiency of photo-electronic conversion and facilitate higher generation of photo-induced charges. Following the orientation of electromagnetic wave oscillation, nonetheless, the light can be categorized into different polarization states. The detection of polarized light is increasingly demanded in various fields, such as optical communications, optical switching, polarization sensors, and spectroscopy, going beyond the traditional methods that focus on the detectivity of light intensity and wavelength in polarization-insensitive photodetectors [94,95,96]. Generally, materials with inherent crystal-structure anisotropy (where atoms exhibit varying arrangements in a 3D or 2D space) can be utilized to observe light polarization direction [97,98,99]. The commonly employed materials include black phosphorus, graphene, and MoS_2_. Unfortunately, the synthesis process of these anisotropic materials is intricate and potentially costly. Precisely engineered optical nanostructures facilitate precise manipulation of incident light and yield distinct electronic signals, implying that special light detection can be performed without the aid of a sophisticated lens array.

This review exclusively focuses on the direct optical manipulation enabled by nanostructured perovskite materials while neglecting the incorporation of metallic or semiconductor meta-surfaces. The simplest optical nanostructure is the grating, which can be employed to recognize linearly polarized light. Su et al. proposed a non-destructive epitaxial growth method to fabricate surface nanopatterned single-crystalline perovskite photodetectors (MAPbBr_3_) and achieve high polarization sensitivity at a specific wavelength. With the simple Au-nanopatterned MAPbBr_3_-Au structure, the polarized light absorption (532 nm) was improved from 87% to 93% (Figure 8a) [100]. Similarly, Wu et al. employed the capillary-bridge confined assembly technique to fabricate highly pure (001)-oriented lead-free chiral 2D double perovskite single-crystalline microwire arrays for enhanced polarized photodetection [101]. The utilization of this confinement growth technology reinforced the crystallization properties of perovskite, yielding the fabrication of highly stable perovskite microwire arrays with exceptional environmental durability. The achievement of an exceptional circularly polarized light photodetector with high responsivity (52 mA W^−1^) and detectivity (3.9 × 10^11^ Jones) was finally realized because of the remarkable crystallinity and sensitive circular-polarization absorption exhibited by chiral perovskite microwire arrays.

Concerning the stability improvement, Park et al. employed the self-assembled block copolymer and nanoimprint strategy to fabricate a 40 nm nanodomains consisting of perovskite crystals with 10 nm in diameter and further form a topological pattern with a periodicity of a few hundred nanometers (Figure 8b) [102]. During the nanoimprinting process, an ultrathin top skin layer was simultaneously formed on the patterned perovskite-copolymer film to ensure that the ordered perovskite can exhibit exceptional environmental stability under ambient conditions for over 100 days. The capillary force with the assistance of a pre-structured template can be also applied to fabricate nanostructured perovskite.

Recently, a capillary-directed self-assembly sequential deposition strategy has been proposed by Taylor and colleagues [98]. When the periodic microchannels are filled with the perovskite solution, it deposits onto the trench of the microstructure, propelled by both capillary forces, and the meniscus of perovskite solution quickly flows through the microfluidic channels (Figure 8c) [103]. Wang et al. established a similar strategy to prepare one-dimensional (1D) organic-inorganic hybrid perovskite nanowires (NWs), simultaneously containing well-defined structures and low crystal defects. The perovskite nanowire photodetector demonstrates exceptional responsivity (1.55 A/W) and detectivity (1.21 × 10^12^ Jones) under the illumination of a 532 nm laser with an intensity of 0.1 μW and a bias voltage of −1 V [104]. Testa et al. have further advanced the microfluidics-assisted technique to enable large-scale and controlled growth of individual perovskite crystals. The performance with a responsivity of 200 AW^−1^ and a rise time of 35 μs can be achieved upon integration into a vertical device featuring a pixelated sensor layer [105].

The high-performance detection of circularly polarized light typically requires the assistance of more intricate optical nanostructures since a simple grating structure fails to meet the necessary criteria. The development of better chiral materials improves circularly polarized light (CPL) detectors based on halide perovskites. Initially, the detection of circularly polarized light (CPL) predominantly relies on the photoelectronic anisotropy of layered structural materials, enabling the accomplishment of single or dual-modal photodetection and imaging capabilities. Nevertheless, the achieved dichroic ratio typically remains below 10 (Figure 9a,b) [106,107,108]. Furthermore, the majority of perovskites employed in CPL detectors have been limited to two-dimensional (2D) structures, which possess optoelectronic characteristics that are incompatible with the prerequisites for exceptional charge transfer properties in directions perpendicular to the plane and broad absorption bands.

By incorporating chiral plasmonic metamaterials or nanoparticles into active perovskite materials, strong light–matter interaction can be obtained through the electromagnetic coupling between the metallic nanostructures and specific wavelengths of light. Joon Hak Oh et al. have incorporated chiral plasmonic gold nanoparticles (AuNPs) into low bandgap 3D mixed Pb-Sn perovskites, allowing for the fabrication of a high-performance CPL detector (Figure 9c) [109,110]. The pre-deposition of chiral metallic nanoparticles onto ground substrates and the prefabrication of chiral plasmonic meta-surfaces through top-down approaches are commonly employed while bringing about compatibility issues with perovskite crystallization. In addition to incorporating chiral optical nanostructures into perovskites, the direct fabrication of intricate perovskite nanostructures for optical manipulation represents the most efficient approach. Furthermore, applying traditional strategies, such as the template-induced self-assembly nanoimprint method, can fabricate more complex optical nanostructures rather than simple grating structures. Notably, Mihi et al. have successfully fabricated CPL-active 2D-chiral perovskite meta-surfaces with a large area through a simple nanoimprint strategy (Figure 9d) [111].

With the advancement of emerging technologies, a growing number of novel strategies are being designed to fabricate intricate perovskite optical structures. For instance, Liu et al. have established a femtosecond laser-induced forward transfer (Fs-LIFT) technology, which can simultaneously realize pattern deposition and alignment of perovskite colloidal quantum dots (QDs) in one single step (Figure 9e) [112]. The salient feature of this strategy lies in its remarkable capability to achieve precise stacked patterning of multiple materials, laying a foundation for the realization of highly flexible optical manipulation involving multi-materials and multi-structures.

**Figure 9 nanomaterials-15-00816-f009:**
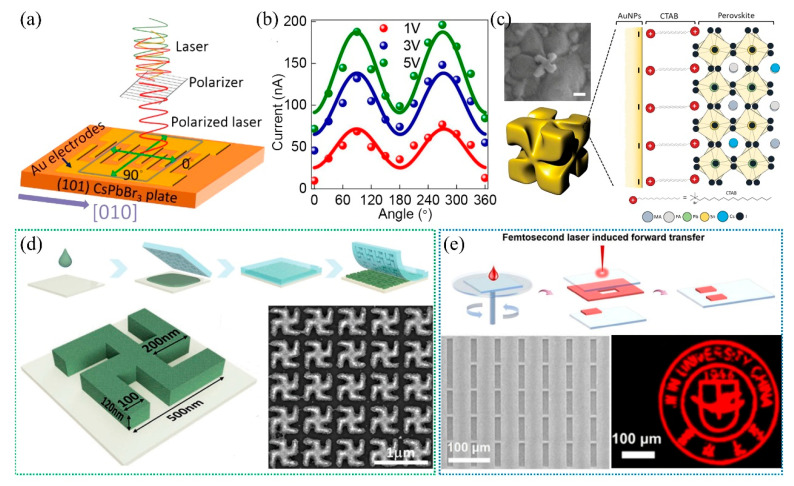
(**a**) Schematic diagram of the planar photodetector and photocurrent measurements on the CsPbBr3 SC under both unpolarized and linearly polarized light. Copyright permissions from Reference [106]. (**b**) Photocurrents of the device under 0.5 mW polarized illumination at various bias voltages. Copyright permissions from Reference [106]. (**c**) Schematic diagram of possible interactions between AuNPs and Cs_0.05_FA_0.5_MA_0.45_Pb_0.5_Sn_0.5_I_3_. Surface SEM micrograph exhibits the morphology of perovskite film prepared separately by a mixed solution of perovskite precursor and AuNPs. The scale bar indicates 100 nm. Copyright permissions from Reference [109]. (**d**) Fabrication of the 2D-chiral meta-surface and schematic diagram of the chiral unit. Large-magnification SEM of an L-gammadion unit is composed of CsPbBr_3_ perovskite NCs. Copyright permissions from Reference [111]. (**e**) Schematic illustration of fabricating patterned red QDs via Fs-LIFT. The SEM image of the irradiated area on the carrier substrates and the fluorescence images of the Jilin University logo. Copyright permissions from Reference [112].

All possible states of incident light polarization by Stokes parameters, which can be geometrically represented on a Poincaré sphere, should be fully characterized to meet the stringent requirements of quantum communication and computation (Figure 10a) [113]. With the advancement of integrated applications, factors such as chiral meta-surfaces and intrinsic chirality of organic molecules are generally adopted to detect circular polarization (ellipticity) by manipulating the spatial distribution of photons possessing distinct spin angular momenta [114]. The detection of linear polarization relies heavily on materials or optical nanostructures with anisotropic dielectric function. Full Stokes photodetection, which involves the determination of all polarization ellipse parameters, typically necessitates the integration of multiple photodetectors with distinct predesigned photonic structures, realizing linear and circular polarization detection in one single device (Figure 10b) [115].

In perovskite materials, the highly crystalline two-dimensional metal halide perovskites possess both self-assembled quantum wells and long-range in-plane exciton transport ability, while the intercalated organic cations could also provide exotic lattice symmetry. The introduction of chiral ammonium compounds enables the breaking of inversion symmetry and facilitates the formation of chiral perovskites to discriminate left (σ^−^) and right (σ^+^) circularly polarized light [116]. Nevertheless, the mere utilization of molecular regulation alone is insufficient to achieve perovskite-based full Stokes photodetection as a result of the relatively low contrast between σ^-^ and σ^+^ polarization photocurrents and the deficiency of linear polarization response.

Combining the predesigned optical nanostructures with the inherent chiral characteristics of the material itself, Wu et al. reported a Stokes parameter photodetector with pre-aligned nanowire arrays of single-crystalline chiral 2D perovskites. While the interlayer chiral cations demonstrate a response to circularly polarized light, the nanowires reveal an anisotropic dielectric function response to linearly polarized light. Finally, light with diverse polarized ellipses is detected by one single device (Figure 10c) [113]. The investigation conducted by Zhao et al. also involves the utilization of the intrinsic chiral optical activity in chiral 2D perovskites and is combined with the integration of another quasi-2D perovskite to form lateral heterojunction nanowire (NW) arrays (Figure 10d) [117]. The anisotropy factor and polarized ratio reach 0.38 and 1.5, respectively. Although the predesigned nanostructured perovskite photodetector can be used to measure all four Stokes parameters, its detectivity still falls significantly short compared to devices integrating complex meta-surfaces with traditional detective materials. More investigations should be performed to achieve high integration and excellent performance of perovskite photodetectors for various specialized applications.

**Figure 10 nanomaterials-15-00816-f010:**
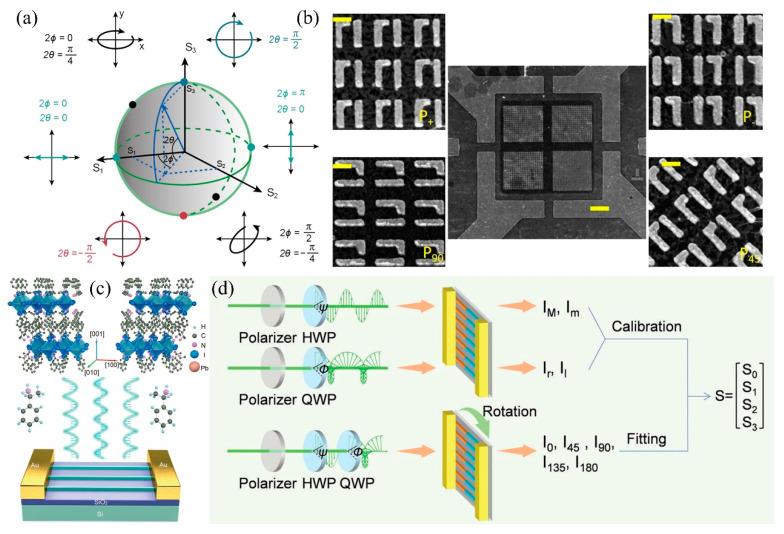
(**a**) Poincaré sphere representation for the polarization states of light with various Stokes parameters. Copyright permissions from Reference [113]. (**b**) SEM images of predesigned optical nanostructures in the polarimeter. Copyright permissions from Reference [115]. (**c**) Crystal structure and device architecture of a Stokes-parameter photodetector. Crystal structures of (S-α-PEA (Phenethylammonium, a organic cations with the formula C_6_H_5_CH_2_CH_2_NH^3+^))_2_PbI_4_ (left) and (R-α-PEA)_2_PbI_4_ (right) chiral 2D-perovskite nanowires. Copyright permissions from Reference [113]. (**d**) Schematic illustration of the experimental designs for full-Stokes polarization measurements. HWP and QWP refer to λ/2 plate and λ/4 plate, respectively. Copyright permissions from Reference [117].

## 5. Conclusions and Perspective

This review provides a concise overview of the utilization of photonic nanostructures for optical manipulation in perovskite photodetectors. Among these strategies, local surface plasmon resonance (LSPR) and surface plasmon polaritons (SPPs), serving as the prevailing phenomena, are commonly employed to effectively manipulate the electromagnetic field on a local scale within the device. The integration of metallic nanostructures into or near the active layer is a crucial step toward achieving electric field regulation in the device. Meanwhile, it should be ensured that their introduction does not compromise the crystal quality of perovskite. Based on this design principle, numerous concise or intricate optical nanostructures have been fabricated and seamlessly integrated into perovskite photodetectors for diverse specialized detection functionalities. Consequently, locally manipulated electric fields in photodetectors offer the advantages of reduced power consumption, improved response speed, minimized dark current, and increased responsivity. In several recent studies, perovskite materials have been directly nanostructured and utilized as optical manipulation elements to reach an elevated level of functional integration. This approach strategy effectively mitigates compatibility issues among different materials, equipping perovskite materials with both light active functionality and optical manipulation capabilities. In this context, the optimization of the crystallization mechanism and the design/fabrication strategy for optical nanostructures have emerged as crucial aspects for advancing research and development in perovskite photodetectors. Nanostructured perovskite enables the manipulation of photons and electrons, presenting a promising approach for enhancing integration and optimizing performance in perovskite photodetectors across various application domains.

## Figures and Tables

**Figure 1 nanomaterials-15-00816-f001:**
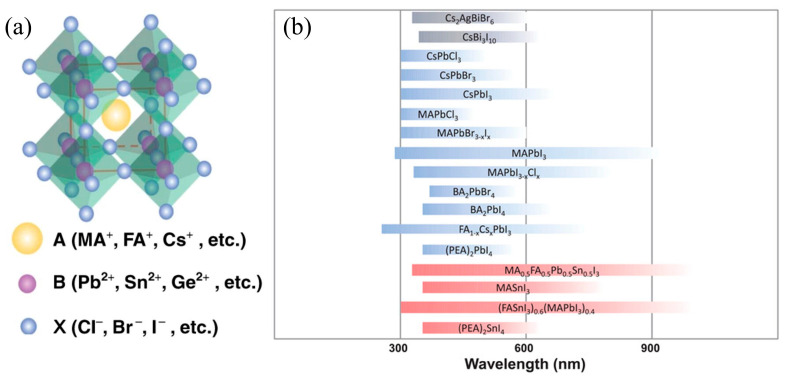
(**a**) Structural features of metal halide perovskite materials. Copyright permissions from Reference [16]. (**b**) Detection ranges for different perovskite photodetectors. Copyright permissions from Reference [20].

**Figure 2 nanomaterials-15-00816-f002:**
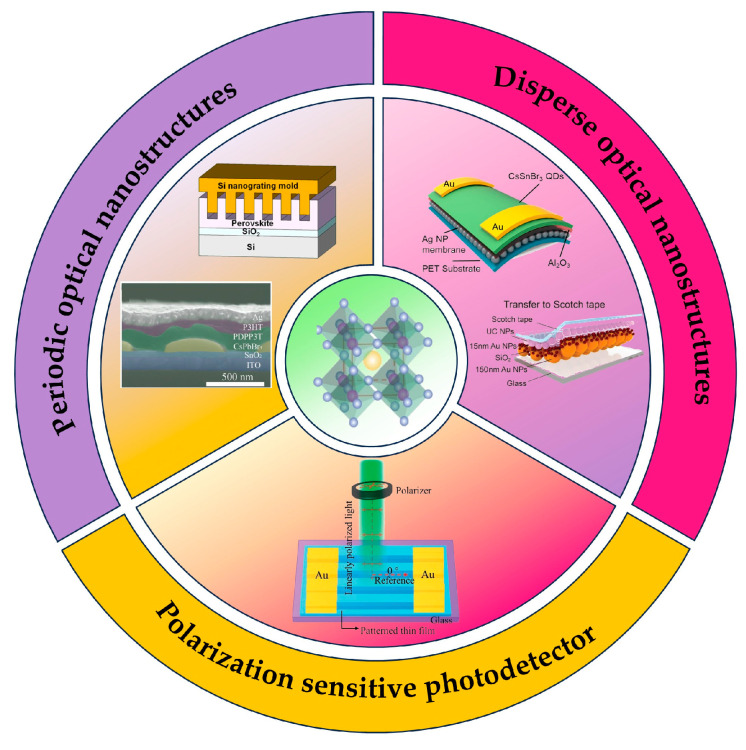
Design of photodetectors based on different optical nanostructures.

**Figure 8 nanomaterials-15-00816-f008:**
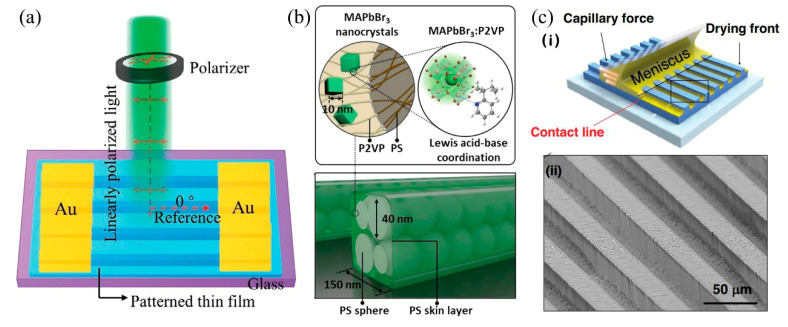
(**a**) Schematic diagram of the photodetector with a MAPbBr3 nanopatterned single crystal device under linearly polarized light. Copyright permissions from Reference [100]. (**b**) Hierarchically ordered MAPbBr_3_/PS-b-P2VP films. Hierarchical structured MAPbBr_3_/PS-b-P2VP was composed of a line pattern of 150 nm in width, PS sphere of ~40 nm in diameter, and MAPbBr_3_ nanocrystals of ~10 nm in diameter. Copyright permissions from Reference [102]. (**c**) The deposition process to fill the patterned PUA substrate with CsPbBr_3_ and SEM image of the tilted side-view of the waveguide channel. Copyright permissions from Reference [103].

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
