# Peer review of "Recent Progress and Future Opportunities for Optical Manipulation in Halide Perovskite Photodetectors"

_nanomaterials, 2025, doi:10.3390/nano15110816_

Round 1

Reviewer 1 Report

Comments and Suggestions for Authors

The manuscript presents a review of various technologies based on optical manipulation, which  have been designed to improve sensitivity or selectivity of perovskite based sensors. The review widely covers various techniques and strategies of optical manipulation with a main focus on metallic nanostructures.

I have few suggestions for improvements:

Figure 1 should have more accurate caption that correspond to the text: there should be X instead of C. The acronyms FA (formamidium), GA(Guanidinium) and PE should be explained.

The unit Grey is rather unusual in connection with the electromagnetic radiation in the wavelength range discussed in the manuscript (optical). It is more common in the ionization radiation.

Plasmonic nanoparticles also absorb light not only create strong optical field, depending on their size. Absorbed radiation can generate heat, which could be detrimental for the detector functionality. It should be commented in the text on page 3. There is also a lower size limit for the NPs for the LSPR effects and enhancement of the optical field is directional. The location of the NPs with regards to the active photosensitive material is therefore important. This should be also commented in the manuscript.

Moreover, plasmonic nanoparticles can also facilitate quenching of excited states in surrounding molecules. This adverse process depends on the distance between the NP surface and an active molecule. It should be commented in the manuscript.

In page 4 there is a note about increased detectivity reached in hybrid  plasmonic upconversion architecture.  First, I think the exact term is specific detectivity and, second, the unit Jones, although well established for this quantity, is not so frequently used and should be explained (cm Hz1/2/ W )

It is written on p. 6 the “In addition to utilizing the hot electrons, harnessing the plasmonic thermal effect holds potential for optimizing perovskite materials“. This sentence is immediately followed, however, with a discussion of the stability issue arising from thermal effects. One would expect here an information how the thermal effects is exploited.

On the page 10 there is a responsivity of 1026.5 A W-1 reported. Does it mean that the accuracy of the device is 0.01 %? It would be really excellent for such kind of device.

General comment: there are many figures taken from the literature showing various phenomena that were exploited for increasing the light harvesting efficiency but the explanatory text is sometimes rather scarce. Better description would be helpful.

Reviewer 2 Report

Comments and Suggestions for Authors

This review article deals with perovskite-based nanostructures for photodetectors, focusing on some light detection techniques.

The topic is interesting. However, the manuscript must be revised, and the following main points must be carefully addressed before it can be considered for publication:

  1. It is important that the authors clarify, inside the manuscript, what is the added value of this review article among the large existing literature on the topic.
  2. The keywords are missing and should be included.
  3. After the names of the two authors there is an "e", perhaps because another author was missing? This should be appropriately corrected.
  4. At the beginning of the introduction, among other structures, it is also worth mentioning alternative systems that compete with perovskites as new engineered detectors. In fact, low-dimensional systems, such as 1D nanostructures, have attracted considerable attention due to their peculiar optical and electronic properties suitable for high performance devices [https://doi.org/10.1063/1.3441404; https://doi.org/10.1021/nl401737u]. On the other hand, 2D quantum wells have shown that their remarkable excitonic properties can be tuned through compositional and structural modifications, with significant consequences on their applications in the field [https://doi.org/10.1103/PhysRevB.50.12179]. However, despite the progress reached with such nanostructures, perovskites represent a highly competitive alternative. Therefore, such discussion, complemented by the suggested articles which are worth mentioning, allow to compare the advantages and performances of different material structures, justifying the choice to focus on perovskite and better understanding the importance of this article.
  5. To improve the readability of the paper, given the size of this review article (18 pages), it is recommended to insert, at the end of the introduction, a short table of contents that shows how the article is organized, to give the reader an overview of its content.
  6. It is necessary that each subsection/paragraph concludes with a brief summary of what has been presented, enriched with critical comments by the authors of this manuscript that highlight the strengths and weaknesses of the various cases presented. In this way, the slavish description of the results of the literature, will be able to benefit from an interesting critical and new vision of the authors.
  7. The conclusions should be completed with a critical view of the overall topic treated, and better and more in depth outline possible future developments/perspectives.
  8. Before the list of References, the “Author contributions” should be added.
  9. Each reference should be completed with the title of the article and the DOI.

Reviewer 3 Report

Comments and Suggestions for Authors

The article entitled “Recent Progress and Future Opportunities for Optical Manipulation in Halide Perovskite Photodetectors” describe a concise overview of the utilization of photonic nanostructures for optical manipulation in perovskite photodetectors.

The results presented in the review are very interesting, the authors have organised the various sections very well and highlighted the key findings. The article is very well written, but I am sure that with the implementation of some suggestions, the clarity and strength of the arguments could be further improved.

  • Line 4 pag 1: The third name is missing among the authors. Check.
  • Line 38-39 pag 1: These two lines are written in bold. Uniform these two lines.
  • Line 49 pag 2: I would suggest that the authors add more recent references on perovskites such as:

1)  https://doi.org/10.1002/adom.202402469

2) https://doi.org/10.1021/acsomega.4c05581

  • Line 60 pag 2: The authors of this cited article are missing.
  • Line 88 pag 3: I would use the word ‘review’ instead of ‘paper’. Replace it in the whole text.
  • Line 172 pag 5: The authors could give references to these works at the end of this sentence.
  • Figure 3: I suggest the authors distance figure 3 from the main text. The layout is not perfect. The text merged with the caption in figure 3. Pay attention to the arrangement of the figures in the text.
  • Line 260 pag 7: Always use the acronym once defined.
  • General comment on the captions of each figure: since the caption of each figure is lengthy, I recommend that you re-read each description carefully.
  • Line 419 pag 11: Insert reference to the work of Taylor et al.
  • Conclusion and perspectives: In this field, what role do low-toxic and lead-free perovskites play, given the growing interest in sustainability in photovoltaic and optical sensing technologies?

Round 2

Reviewer 1 Report

Comments and Suggestions for Authors

The authors responded to most of the objections and have markedly improved the text. I agree with their answers besides the second from the end, concerning the accuracy. Any value should be shown with the number of valid digits as it corresponds to the accuracy of the measurement. I doubt that this basic rule has been respected.

In this case, the authors did not respond to my comments adequately.

Reviewer 2 Report

Comments and Suggestions for Authors

The authors have addressed the issues raised and have thus improved the article. However, there are still the following issues that needs to be corrected:

  1. Ref [10]: the title of the paper is wrongly reported, the compound mentioned in the title needs to be corrected as “Zn1-xCdxSe/ZnSe”,
  2. Ref [30]: the title of the paper is wrongly reported; the compound mentioned in the title needs to be corrected as “Cs2PtxSn1−xCl6”.

I suggest checking in the article proofs all the references to be sure that they are correctly reported. I am confident that the authors will do so.

Author Response

请参阅附件。
